# Recombinant Peptide Production Softens *Escherichia coli* Cells and Increases Their Size during C-Limited Fed-Batch Cultivation

**DOI:** 10.3390/ijms24032641

**Published:** 2023-01-30

**Authors:** Andreas Weber, Martin Gibisch, Daniel Tyrakowski, Monika Cserjan-Puschmann, José L. Toca-Herrera, Gerald Striedner

**Affiliations:** 1Christian Doppler Laboratory for Production of Next-Level Biopharmaceuticals in E. coli, Institute of Bioprocess Science and Engineering, Department of Biotechnology, University of Natural Resources and Life Sciences, Muthgasse 18, 1190 Vienna, Austria; 2Institute of Biophysics, Department of Bionanosciences, University of Natural Resources and Life Sciences, Muthgasse 11, 1190 Vienna, Austria

**Keywords:** bacteria, peptide expression, membrane properties, periplasmic space, viscoelasticity, atomic force microscopy

## Abstract

Stress-associated changes in the mechanical properties at the single-cell level of *Escherichia coli* (*E. coli*) cultures in bioreactors are still poorly investigated. In our study, we compared peptide-producing and non-producing BL21(DE3) cells in a fed-batch cultivation with tightly controlled process parameters. The cell growth, peptide content, and cell lysis were analysed, and changes in the mechanical properties were investigated using atomic force microscopy. Recombinant-tagged somatostatin-28 was expressed as soluble up to 197 ± 11 mg g^−1^. The length of both cultivated strains increased throughout the cultivation by up to 17.6%, with nearly constant diameters. The peptide-producing cells were significantly softer than the non-producers throughout the cultivation, and respective Young’s moduli decreased by up to 57% over time. A minimum Young’s modulus of 1.6 MPa was observed after 23 h of the fed-batch. Furthermore, an analysis of the viscoelastic properties revealed that peptide-producing BL21(DE3) appeared more fluid-like and softer than the non-producing reference. For the first time, we provide evidence that the physical properties (i.e., the mechanical properties) on the single-cell level are significantly influenced by the metabolic burden imposed by the recombinant peptide expression and C-limitation in bioreactors.

## 1. Introduction

Bacterial cells are capable of growing in a wide range of different environments and are constantly enduring physical challenges from their surroundings. These include osmotic/turgor pressure variation, shear flow, and shape changes during growth and cell division [1]. These challenges are withstood by the bacterial cell envelope, a complex network of different macromolecules such as proteins, lipids, and (lipo)polysaccharides [2]. As part of the cell envelope, the cell wall has important functions related to the cell shape, cell division and growth, adhesion, motility, and exchange of nutrients and metabolites [3,4]. Little is known about the connection of bacterial mechanics to their function, such as the influence of metabolic burden, the role of different cytoskeletal and periplasmic proteins, and the mechanisms involving growth and shape changes during division [5,6,7]. In gram-negative cells, such as *Escherichia coli*, the cell wall is made up of the inner membrane (IM), a periplasmic space that contains a thin peptidoglycan (PG) layer, and the outer membrane (OM) [8,9,10]. A diverse set of lipoproteins maintain the connection of the OM to the peptidoglycan layer, which is believed to provide mechanical integrity to the bacterial cell. Recently, the mechanical role of the outer membrane was described [11,12]. The cell wall withstands the pressure inside the bacteria (the turgor pressure), which is in the range of a few atm [13,14].

*E. coli* is extensively used as the expression host for the large-scale, rapid, and cost-effective production of recombinant proteins and peptides (<100 amino acids). Many recombinant gene products, however, can impose adverse effects on the host in addition to metabolic burden [15]. Thus, process optimization is often needed to favor feasible protein/peptide production. Efforts for process optimization, include periplasmic production for disulphide bond formation, a reduced cultivation temperature for increased folding mechanics, integration of the expression cassette into the genome for stress reduction, and many more [16,17]. Most optimization efforts predominantly focus on enhanced process performance (product yield) and changes at the physiological level. The impact of the recombinant protein or peptide expression on physical properties (i.e., the shape, mechanics, and structural changes in cell wall architecture) of the host organism remains an open question.

The mechanical properties of single bacterial cells have been investigated by a number of different methods. These techniques include atomic force microscopy (AFM), osmotic perturbations in parallel with optical imaging or scattering techniques, microfluidic techniques, encasement in gels, mechanical modelling, and optical tweezers [4,18,19,20]. Bacteria exhibit Young’s modulus (E) values in the range of a few MPa, a stiffness of around 0.001 to 1 N m^−1^, and turgor pressures of a few atm [18,21]. AFM-based indentation studies are widely used for measuring bacterial mechanics, as they allow precise indentation, time- and frequency-dependent measurements, and combination with optical and spectroscopic techniques [22,23]. Most studies on bacterial mechanics have been performed with overnight batch cultivations without defined control of growth rates, oxygen, and nutrient supply [24,25,26].

During recombinant protein or peptide production, part of the energy available to the bacterium is used for synthesis, folding, and protein transport. Resources for cell maintenance and growth are, therefore, rapidly depleted, and the host organism can be overburdened, especially when powerful expression systems (e.g., DE3-derived T7) are used. The rise in turgor pressure associated with increased amounts of intracellular protein could even further increase the stress on the host. We hypothesize that this additional metabolic load can lead to failures in maintaining the structural integrity of the cells, with consequent changes in the cell shape, size, and mechanical properties. We therefore set out to study the influence of peptide production on the mechanical and morphological properties of *E. coli* over time. This was implemented by a carbon-limited exponential fed-batch cultivation with a defined medium and precisely controlled process parameters (growth rate, temperature, dissolved oxygen, and pH). Samples were drawn throughout the cultivation, and AFM-based imaging and force spectroscopy were performed. We present mechanical time-course data on the single-cell-level of bacteria derived from a defined production process in bioreactors. Recombinant peptide production leads to significant softening of cells, accompanied by an increase in cell volume. Moreover, non-producing cells also undergo changes in mechanical properties, probably caused by prolonged carbon limitation in the exponential feed phase. This study strongly suggests a connection between the metabolic burden of recombinant peptide production, carbon-limited conditions, and the structural integrity of bacterial cells.

## 2. Results

### 2.1. Establishment and Evaluation of AFM Sample Preparation Protocol

To ensure comparability and interpretation of the AFM data, the availability of a robust and reproducible methodology for sample processing is a prerequisite. Of particular importance was the evaluation of whether and to what extent of storage of the cells by freezing would affect the AFM measurement and what variation can be expected in each sample’s preparation protocol.

Therefore, AFM measurements with freshly harvested and frozen/thawed cells from induced and non-induced shake flask cultivations with a Fab-producing *E. coli* BL21(DE3) strain, referred to as B <oFTN2>, were carried out. Biological triplicates of the induced Fab-producing cells showed no significant differences, with an average Young’s moduli of 2.73 ± 0.75, 2.73 ± 0.87, and 2.87 ± 0.60 MPa (see Figure 1). Similar results were obtained for the measurements with freshly harvested and frozen-induced samples, with a Young’s modulus of 2.66 ± 0.94 MPa. For the non-induced samples, a YM of 2.75 ± 1.18 MPa and 2.53 ± 0.76 MPa for the fresh and frozen samples were determined, respectively. For samples producing CASP-SST, the fresh samples had an average YM of 1.32 ± 0.80 MPa, while the frozen samples delivered one of 1.27 ± 0.65 MPa. This confirms (a) that our sample preparation method provides reproducible results with similar measurement errors and (b) that freezing/thawing does not influence AFM measurements. We could, therefore, study the impact of peptide expression on mechanical and viscoelastic properties using these established methods for sampling and AFM.

### 2.2. Bioreactor Cultivations of Producing and Non-Producing Strains

The influence of recombinant peptide expression and generally stressful conditions on mechanical properties of single cells during carbon-limited fed-batch cultivation in bioreactors was studied. We cultivated a peptide-producing and a non-producing (reference) BL21(DE3) host strain, hereafter referred to as *PEP* and *REF*, respectively. During the initial hours of induced feed, REF and PEP grew as theoretically predicted (Figure 2A). While the REF culture remained growing, as precalculated throughout the cultivation, the PEP culture deviated from REF and showed impaired growth kinetics, especially from 16 h on. Surprisingly, PEP appeared to recover from initially lower (actual) growth rates (Figure 2C) between 19 and 23 h, exceeding the growth rates of REF. This inversed after 23 h when PEP showed a decreasing µ of between 23 and 27 h. The REF cultures showed higher growth rates than the PEP cultures; however, a subtle drop appeared between 19 and 23 h, with a constant µ from then on. Residual glucose was exemplarily analysed for PEP using HPLC to confirm carbon limitation (data is supplemented in Appendix A). The residual glucose content specific to the respective biomass was found to be 2.2% on average throughout the cultivation, which corresponds to roughly 1.4% of fed glucose at respective time points. A minor accumulation was detected, which was considered negligible.

Recombinant CASP-SST was expressed in considerable amounts during fed-batch (Figure 2B) without any inclusion body formation (not shown). After 19 h of feed, the specific peptide content reached its maximum of 197 ± 9 mg g^−1^; however, it decreased by over 50% onwards until 31 h. As shown in Figure 2D, PEP strongly lysed between 11 and 23 h up to 18.2%; however, the percentual cell lysis decreased from then on to roughly 8% after 31 h. REF showed relatively constant levels of cell lysis at around 7.4% throughout the cultivation, indicating a lower stress level.

### 2.3. Influence of Peptide Production on Cell Morphology

Samples of REF and PEP were imaged with AFM in QI mode. Figure 3 shows representative 3D height images corresponding to culture times of 3 h (before induction), 19 h, and 31 h. After 3 h, the bacteria length, diameter, and volume appeared to be similar, while longer growth times led to larger cell lengths. This effect was more pronounced in PEP. In addition, for PEP, nanometric protrusions were seen on the surface of the outer membrane. We found no significant changes in cell structure, such as holes or defects in the cell wall.

The results of the cell length, diameter, and volume analysis can be seen in Figure 4 (details are shown in Appendix A). For both cultures, the length of the bacteria increased significantly over the course of the C-limited cultivation. For REF, an increase from around 1.65 µm after 3 h to 1.9 µm after 31 h was observed. This increase was more pronounced in PEP, as their length started at 1.7 µm, rising to 2 µm after 27 h. For REF, the increase in length appeared as one step between 19 and 23 h, while for PEP, the increase happened gradually. The diameters of REF and PEP showed no significant differences in comparison; however, PEP increased in diameter after 19 h and gradually decreased again until 31 h. Similar observations could be made for REF; however, the decrease in diameter was more distinct after 23 h. Both strains had a starting volume of around 0.9 µm^3^, which remained similar for REF over the first 19 h of cultivation, to increase to 1.25 µm^3^ after 23 h and then decrease again. For PEP, the volume increased to around 1.3 µm^3^ after 19 h already and appeared to reach a plateau there, with only small decreases to the final values of 1.2 µm^3^. The same behaviour can be seen for the surface area. The surface area to volume ratio remained constant for both samples. Changes in the volume and surface area can be underlined by a variation in length and diameter.

### 2.4. Influence of Peptide Production on Mechanical Properties of Bacterial Cells

We used AFM measurements to determine the mechanical properties of the cells (Figure 5A). For a linear elastic material, deformation is proportional to the applied stress and is described by Young’s modulus (see Equation (3), often used synonymously for stiffness). The maximum indentation, δ, at the highest load was similar for both REF and PEP at the starting point, being 40 and 37 nm, respectively. For REF, this value stayed approximately constant, and only a significant drop in the indentation value was seen at 27 h (32 nm). For PEP, an increased indentation to a maximum of 100 nm after 23 h was observed, with receding indentations of 92 and 76 nm for 27 and 31 h, respectively. As the increase in deformation is proportional to a lower stiffness, these findings indicate that the cells become softer due to peptide production.

As expected, an inverse behaviour for the apparent Young’s modulus, *E*, to the indentation can be seen (Figure 5B). The starting values (corresponding to the non-induced cells) were 3.1 MPa for the reference cells and 3.8 MPa for the PEP. Throughout the cultivation, Young’s modulus increased significantly for the reference cells, with a maximum of 4.0 MPa after 27 h (and 3.7 MPa after 31 h). PEP showed a drastic decrease in cell stiffness, reaching a minimum after 23 h of fermentation with a Young’s modulus value of 1.6 MPa (50% of reference). The elastic modulus then slightly increased to the final value of 2.3 MPa.

### 2.5. Peptide-Producing Cells Appear Less Solid-Like

Biological materials exhibit a complex hierarchical architecture, with the components showing different orders of size, organisation, and relaxation behaviour. Therefore, viscoelastic models are best suited to reflect their mechanical properties, which were investigated by performing creep measurements. The measurements showed a monotonically increasing creep function that was modelled using a power law approach (creep curve fitting is shown in Appendix A). This approach fitted the data better than the standard linear solid model (see supporting information). We derived the indentation at the beginning of the creep phase (*δ*_0_), creep magnitude (*δ_t_*), power law exponent (*β*), and the relaxation modulus (*E*_0_). The creep magnitude is the overall deformation during the creep phase (the larger it is, the softer the material is), the power law exponent indicates how solid- or liquid-like a material behaves, and the relaxation modulus is an indicator of sample stiffness.

Indentation at the beginning of the creep phase showed a similar behavior compared to the one measured in the elasticity experiments (Figure 6). The creep response (*δ_t_*) remained comparable throughout the cultivation for REF and increased from an initial 20 nm to around 60 nm for PEP. *E*_0_, the relaxation modulus, was initially higher for the peptide-producing cells (2 MPa against 1 MPa for the reference ones), to become lower over time (0.56 MPa for the reference and 0.28 MPa for the producing samples). For the reference cells, a peak in the relaxation modulus was reached after 23 h. Softening of the peptide-producing cells was accompanied by fluidisation. The power law exponent, *β*, increased from the initial values of 0.03 to 0.08, while it stayed nearly constant for the reference.

## 3. Discussion

Cell morphology changes due to carbon limitation, even under constant growth conditions in a tightly controlled bioreactor environment. In addition, we provide evidence that recombinant peptide production leads to significant softening and fluidisation of *E. coli* cells. Previous studies regarding mechanical properties and morphology of bacteria mostly used small-scale cultivation in shake flasks with low cell densities, and the cells were usually harvested after a short time of growth. The results obtained in this study revealed that the shape and mechanical properties of the cells contain process-relevant information. They presumably can contribute to the improvement of the design and characterization of future bioprocesses.

Intracellular protein accumulation is a result of the interplay between synthesis and degradation [27]. Our results imply a specific peptide maximum after approximately 19 h of cultivation, with a decreasing peptide production from then on. Before the maximum, the ratio between the synthesis and degradation was on the synthesis side, which seemingly inversed afterwards. Fast-growing bacteria such as *E. coli* are known for fast adaptation towards various extra- and intracellular changes. We suggest that regulatory adaption mechanisms negatively influence recombinant peptide expression [28]. Moreover, a considerable number of cells in the performed cultivations lysed, likely due to overloading the cells with the strong T7 promoter system [15]. This led to increasing amounts of the extracellular peptide. The influence of product formation on the biomass yield and glucose consumption under described cultivation conditions has been characterized in a detailed C-balancing study [29]. Deviations in the biomass of PEP, in comparison to the theoretical growth curve, can, therefore, be ascribed to shifts in the metabolism toward an increased formation of CO_2_ and acetate formation.

The volume and surface area, like the cell shape, are regulated by complex molecular machinery and are important for growth, motility, and nutrient uptake [30]. Bacteria show a constant surface-to-volume scaling law that can be written as S=γV23, where *γ* is a constant prefactor depending on the shape and growth state of the cell. Schaechter’s empirical law connects the cell volume and growth rate as $V=V_{0}e^{\alpha \kappa }$, where $V_{0}$ is the volume at the zero-growth rate, α is the rate of increase, and κ is the population growth rate [31,32]. As shown in this study, an increase in volume was always accompanied by a scaled increase in surface area, confirming our expectations regarding the scaling prediction (see Appendix A). Surface area-to-volume homeostasis is either a result of a constant proportion between the rate of cell elongation and the accumulation of proteins responsible for cell division or regulated by peptidoglycan (PG) synthesis [7,33]. The reasons for cell elongation without separation are not clear; however, a variety of proteins are essential for septal PG splitting and daughter cell separation [34]. Consequently, the effects on cell–cell separation machinery caused by recombinant peptide expression or C-limitation cannot be neglected. The influence of product formation on the biomass increases, and glucose consumption, under similar cultivation conditions, has been characterized in detailed C-balancing studies [35]. Based on these results, the deviation of the achieved biomass in the PEP experiment from the theoretical value can be attributed to shifts in the metabolism leading to increased CO_2_ and acetate formation.

In a simplified way, rod-shaped bacterial cells can be considered as anisotropic balloon structures, where their shape depends on the elasticity of the cell wall, which is stretched by the internal pressure that is larger than the outside pressure (the positive turgor pressure). The elasticity of the cell wall is mainly defined by the peptidoglycan layer in the periplasmic space and the pre-stressed outer membrane, as recently shown [11]. The production of recombinant human somatostatin-28 fused to the CASPON tag leads to significant softening of *E. coli* cells shortly after induction, which is accompanied by an increase in cell volume and length. This is the result of the cells being forced to invest energy in recombinant peptide expression that would otherwise be used for regular metabolic functions, such as growth, division, and motility [15]. Consequently, less energy was invested in these essential physiological functions, and the metabolic burden could result in a lack of cell wall synthesis. Other to date unknown and unpredictable interactions of recombinant peptides/proteins with the production host could further negatively affect the cell. Moreover, adverse effects caused by highly negative tags (such as the CASPON tag) cannot be neglected and could interact with the negatively charged outer membrane but also influence the host cell on a global level. More investigations of changes in the expression levels of proteins with functions in the structure, shape, and bacterial mechanics as a result of recombinant protein expression are needed [6].

After inducing protein production, the product accumulates in the cell or in the periplasmic space, leading to higher turgor pressures [35,36]. The constant elasticity of the surrounding cell wall will lead to an increase in the volume. AFM force spectroscopy measurements enable the estimation of turgor pressures from the indentation curves (the linear region after the elasticity region), analytical modelling, and creep curves. Estimated turgor pressures and elastic properties change at similar timescales [13]. However, we have omitted, in detail, the analysis here, as the separation of the elastic and turgor pressure effects is not trivial. By the determination of viscoelastic moduli from the creep curves, a softening of the producing cells with parallel fluidisation was shown.

An important aspect of our results is the change in the mechanical properties over time. For both analysed strains, the changes in the mechanical properties are connected to the volume and morphology. Additionally, intracellular accumulation of the recombinant peptides, decreased state of health, and the resulting tendency for cell lysis contributed to the mechanical properties. An interesting counter-intuitive effect was observed for the non-producing culture: The elasticity seemed to increase over the 31-h fed batch cultivation, while the cell volume showed a peak at 23 h and then decreased. The elasticity appears to have increased over the 31-h long fed batch, while the cell volume showed a peak at 23 h and later decreased (the Young’s moduli of various batch cultures is shown in Appendix A). We show that the cells could potentially undergo certain changes on the single-cell level caused by carbon-limited conditions. Even without the impact of recombinant peptide expression-related stress, C-limitation can change various intracellular processes that could affect the organism on a global level [37]. For peptide-producing cells, the maximum volume observations at 19 h coincide with the softest cells, as well as the highest intracellular protein concentration and cell lysis. This is supported by a recent study showing that the over-expression of a non-functional protein leads to a significant increase in volume in *E. coli*. The authors show that the density of individual bacterial cells appears to be conserved, hinting towards a similar turgor pressure and that the cell wall elasticity is the main factor for changes in the bacterial mechanics because of the protein expression [38].

In summary, we have provided evidence that metabolic burdens affect the mechanical properties, shape, and volume of bacterial cells. Not only metabolic stress due to recombinant peptide production but also growth under C-limited conditions have an impact on the host organism. Our results underline the importance of considering well-defined process conditions when investigating the relationship between the mechanical properties of bacteria and their growth. With respect to future studies, we can envision that data on mechanical properties and structural integrity could be parameters to consider in bacterial bioprocesses.

## 4. Materials and Methods

### 4.1. Strains

BL21(DE3) (New England Biolabs, Ispwich, MA, USA) and two BL21(DE3)-derived strains for peptide, BL21(DE3)<oCASP-SST>, and antibody fragment (Fab) production, (BL21(DE3)<oFTN2>), were used in this study. Human somatostatin-28 (SST, 3.2 kDa), fused to the N-terminal CASPON tag (4.2 kDa) [39], as well as a Fab specific to the tumour necrosis factor, α (FTN2, 47.2 kDa [16]), were expressed. Both peptide and Fab were expressed with an N-terminal OmpA signal sequence for translocation into the periplasmic space for disulphide bond formation. The expression cassettes under the control of the *lacUV5* promoter and the tZENIT terminator [40] were integrated into the genome of BL21(DE3) at the attTn7 site using a previously described method [41].

### 4.2. Media

For cultivations in shake flasks, a semisynthetic minimal medium (SSM) was used and, per litre, contained 3 g KH_2_PO_4_, 4.58 g K_2_HPO_4_, 0.1 g tryptone, 0.05 g yeast extract, 0.25 g tri-sodium citrate, 0.1 g MgSo_4_*7H_2_O, 0.01 g CaCl_2_*2H_2_O 150 µL trace element solution (see below), 0.45 g (NH_4_)_2_SO_4_, 0.37 g NH_4_Cl, and 3.3 g glucose monohydrate. A bioreactor medium was based on the SSM and was individually designed for the batch and fed-batch phases. The batch medium was planned to harbour 2 g of biomass in 500 mL and contained 8.5 mg L^−1^ KH_2_PO_4_, 2.9 mg L^−1^ 85% H_3_PO_4_, 0.6 mg L^−1^ yeast extract, 3.7 mg L^−1^ tri-sodium citrate, 0.18 mg L^−1^ MgCl_2_*H_2_O, 0.08 mg L^−1^ CaCl_2_*2H_2_O, 0.2 mg L^−1^ (NH_4_)_2_SO_4_, 4.1 µL L^−1^ of a trace element solution, and 13.2 g L^−1^ glucose monohydrate. Designed for 43 g of biomass in 750 mL, the feed medium contained 2.7 g L^−1^ MgCl_2_*2H_2_O, 1.2 g L^−1^ CaCl_2_*2H_2_O, 2.9 µL L^−1^ of a trace element solution, and 193 g L^−1^ glucose monohydrate. The trace element solution contained FeSO_4_*7H_2_O (40 g L^−1^), MnSO_4_*H_2_O (10 g L^−1^), AlCl_3_*6H_2_O (10 g L^−1^), CoCl_2_*7H_2_O (7.3 g L^−1^), ZnSO_4_*7H_2_O (2 g L^−1^), Na_2_MoO_4_*2H_2_O (2 g L^−1^), CuCl_2_*2H_2_O (1 g L^−1^), and H_3_BO_3_ (0.5 g L^−1^).

### 4.3. Cultivations

Shake flask experiments were performed in SSM, inoculated with an OD_600_ of 0.5, and induced with 0.5 mM Isopropyl β-D-1-thiogalactopyranoside (IPTG) after 2 h. The production phase lasted 4 h, and samples for AFM were drawn at the end of cultivation. For bioreactor cultivations, DASGIP^®^ SR1500ODLS benchtop bioreactors (Eppendorf AG, Hamburg, Germany) were used with respective hardware for online monitoring and controlling. The pH was maintained at 7 and controlled by supplementation of ammonia. Bioreactors were inoculated with ~7 mg of biomass. Precultures in shake flasks were grown in SSM at 37 °C while shaking at 200 rpm. After the batch phase, an exponential fed-batch with a specific growth rate, µ, of 0.1 h^−1^ was initiated and maintained for 31 h. Expression of CASPON-SST was induced after 3 h of feed by the pulsed addition of 89 µmol IPTG (174 µM) at the induction timepoint. Temperatures for batch and fed-batch phases were 37 and 30 °C, respectively.

### 4.4. Analytics

Samples were taken throughout cultivation to determine the dry cell mass (DCM), as well as the intracellular (IC) and extracellular (EC) peptide content. For AFM measurements, cell suspension samples were diluted 1:2 in 80% glycerol and stored at −80 °C until further use. For peptide quantification, cell pellets containing 1 mg of DCM were resuspended in a lysis buffer (27 mM Tris/HCl, 25 mM EDTA, 10 mM MgCl_2_*6H_2_O, and 0.8% of a reducing agent (Invitrogen, Waltham, MA, USA)) and lysed via sonication. Peptide content was analysed via Tricine SDS-PAGE (see Appendix A) [42] using Novex^TM^ 10–20% Tricine mini gels and respective reagents (Invitrogen^TM^, Waltham, MA, USA). ImageQuant^TM^ (Cytiva, Marlborough, MA, USA) was used for the quantification of peptide bands via bovine serum albumin (BSA) standards. DNA content in the cell-free cultivation supernatant was analysed using a Qubit 4 Fluorometer (Thermo Fisher Scientific, Waltham, MA, USA) for estimation of cell lysis. Percentual cell lysis was estimated based on the assumption that 1 g of biomass contains approximately 20 mg of DNA [43], and that extracellular DNA represents damaged/lysed cells [44,45]. Cell lysis quantification is shown in Equation (1) (growth and cell lysis curves are shown in Appendix A).
(1)$$Percentual\;cell\;lysis\;\left[\%\right]=\frac{{\left[DNA\right]}_{extracell.}}{{\left[DNA\right]}_{extracell}.+\left(DCM*20\;mg\;g^{-1}\right)}*100$$

### 4.5. AFM Sample Preparation

Bacterial suspensions were thawed and centrifuged for 5 min at 12,000 rpm. The pellet was washed three times in PBS (phosphate-buffered saline, pH 7.4). Borosilicate glass slides (24 mm diameter, 0.1 mm thickness, Menzel Gläser, Braunschweig, Germany) were functionalised with 0.2% PEI at room temperature overnight. This was followed by rinsing the glass slides three times with PBS, and 1 mL of the bacterial suspension was added. Bacteria were left to adhere to the surface (due to electrostatics) for 1 h, and then the sample was washed and transferred to the AFM.

### 4.6. AFM Setup

For AFM measurements, a JPK Nanowizard III system (Bruker, Berlin, Germany) was used to perform imaging, force spectroscopy, and creep measurement. The AFM system was built onto an inverse optical microscope (Axio Observer Z1, Zeiss, Berlin, Germany), and a temperature-controlled liquid measurement chamber was used. Triangular MSCT cantilevers E with pyramidal tips (a nominal tip radius of 10 nm, resonance frequency of 38 kHz in the air, and a spring constant of 0.1 N/m; Bruker, Germany) were used. Prior to and after measurements, they were cleaned with EtOH, acetone, and 30 min periods of UV/O cleaning. Sensitivity and the spring constant were calibrated by the thermal tune method, making use of the equipartition theorem. The system was left to equilibrate 30 min prior to measurements. Each sample was produced in triplicates and measured for a maximum of three hours.

### 4.7. Bacterial Imaging and Morphological Property Determination

Bacteria were imaged using the Quantitative Imaging mode developed by JPK (Bruker, Germany). Square areas of 15 × 15 µm^2^ up to 20 × 20 µm^2^ were defined with a resolution of 256 × 256 pixels. Measurements were performed with a curve length of 1 µm, a force setpoint of 0.3 nN, approach and retract rates of 100 µm s^−1^, and sampling rates of 200 kHz. At least three images were performed per sample. Image processing was done in the JPKSPM software (JPK/Bruker, Berlin, Germany version 6.1.195). In addition, the bacterial length and diameter were measured with the JPKSPM software. The volume and surface area of bacteria was approximated by modelling the bacteria as a cylinder encased by two half spheres.

### 4.8. Force Spectroscopy and Creep Measurements

After the position of the bacteria was determined by QI imaging, force spectroscopy and creep measurements were performed. Regarding the prior, an array of 8 × 8 curves was measured over an area of 250 × 250 nm^2^ in the central region of the bacteria with an approach speed of 10 µm s^−1^ and a maximum force of 0.6 nN. A curve length of 1 µm and a sampling rate of 20 kHz were used. At least 50 bacteria per sample were measured. For creep measurements, an additional constant force (at 0.6 nN) segment for 2 s was added. An array of 2 × 2 measurements per cell with at least 20 cells per sample was measured.

### 4.9. Data Analysis

Analysis of force–distance and indentation–time curves were performed using an R afm Toolkit [46]. The curves were extracted using the JPKSPM software, imported to the toolkit, and then contact and detachment points were determined using an algorithm already described before with an optimized set of parameters [47]. After this, a baseline correction was performed, the zero-force point was determined, and the deformation of the sample, $\delta_{s}$, was calculated as follows
(2)$$\delta_{s}=Z_{p}-\delta_{c}=Z_{p}-\frac{F}{k_{c}},$$
with the cantilever as an ideal Hookean spring of stiffness, $k_{c}$, the z-position of the piezo as $Z_{p}$, the applied force as F, and the deformation of the sample as $\delta_{s}$ (later only denoted as δ) [48]. The initial non-linear indentation segment (20 nm) of the force–distance curves was fitted by the Sneddon extension of the Hertz model for linear elastic materials as used for a pyramidal indenter as
(3)$$F=\frac{E_{app}}{1-\nu^{2}}\frac{\tan\alpha }{\sqrt{2}}\delta^{2},$$
where $E_{app}$ is the apparent Young’s modulus, ν is the Poisson’s ratio of the material (set as 0.5), and α is the face angle of the pyramid of 17.5 °. Values for each bacterial cell were pooled, and a minimum fitting quality of R^2^ of 0.9 was defined. The change of indentation with time (creep curves) for an increase in the contact area can be expressed as [49,50]
(4)$$\delta^{n}\left(t,F\left(t\right)\right)=\frac{1}{C_{n}}\int_{0}^{t}J\left(t-\xi \right)\frac{\partial \delta^{n}}{\partial \xi }d\xi ,$$
where n and $C_{n}$ depend on the indenter geometry (n=2 for the pyramidal tips, $C_{n}=\frac{1}{1-\nu^{2}}\frac{\tan\alpha }{\sqrt{2}}$), J(t) is the creep compliance, and ξ is a dummy time variable [51,52]. We assumed an instantaneous ramp leading to a straightforward solution for the creep segment as
(5)$$\delta^{n}\left(t\right)=\frac{1}{C_{n}}J\left(t\right)F_{0}.$$

Here, $F_{0}$ is the hold force. For the creep compliance, J(t), of the viscoelastic bodies, different models have been used in the literature, such as the standard linear solid (one spring in series with a Kelvin–Voigt element) or a power law rheological model. The latter was used as
(6)$$J\left(t\right)=\frac{1}{\Gamma \left(1-\alpha \right)\Gamma \left(1+\alpha \right)}\frac{t^{\alpha }}{E_{0}},$$
where Γ is the gamma function, α is a power law exponent (0<α<1, with 0 being an ideal elastic solid and 1 being an ideal viscous liquid), and $E_{0}$ is the relaxation modulus (at a time of 1 s used as the scale factor) [53,54]. A higher power law exponent indicates that the material behaves more liquid-like, while a lower one indicates a more solid-like material. In addition, a standard linear solid model in the Kelvin–Voigt representation was used as
(7)$$J\left(t\right)=\frac{1}{E_{1}}+\frac{1}{E_{2}}\left(1-e^{-t\frac{E_{2}}{\eta_{2}}}\right),$$
with $E_{1}$ as the instantaneous response, $E_{2}$ as the delayed response, and τ as the characteristic response time (viscosity is defined by $\eta_{2}=E_{2}\tau_{2}$) [55].

### 4.10. Statistics

Measurements were pooled, and properties were calculated, as described above. Statistical analysis was performed in Origin Pro 2018 (OriginLab, Northampton, MA, USA). For normally distributed samples, ANOVAs were performed, and for not-normally distributed samples, Kruskal–Wallis ANOVAs were performed. Significances are indicated in the figures and reported as “*” for *p* < 0.05, “**” for *p* < 0.01, and “***” for *p* < 0.001.

## Figures and Tables

**Figure 1 ijms-24-02641-f001:**
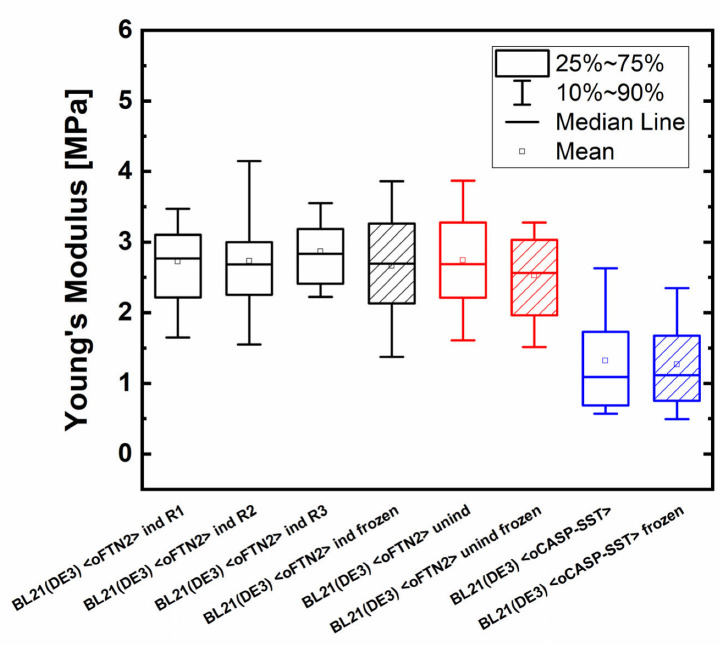
Young’s modulus comparison of freshly harvested or frozen/thawed BL21(DE3) <oFTN2> samples and BL21(DE3) <oCASP-SST. The number of measured cells was 50, 27, 26, 30, 49, 61, 52, and 39 respectively. Black boxes: induced BL21(DE3) <oFTN2> cells, red boxes: uninduced BL21(DE3) <oFTN2> cells, blue boxes: induced BL21(DE3) <oCASP-SST> cells.

**Figure 2 ijms-24-02641-f002:**
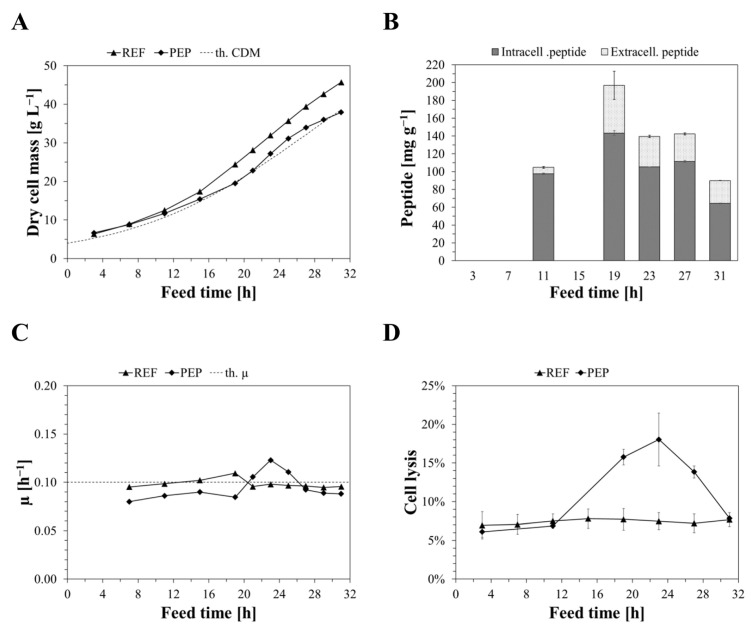
Growth curves of strains of REF and PEP in comparison to the theoretical DCM (th. DCM) (**A**), analysis of intracellular and extracellular peptide content specific to biomass (**B**), actual specific growth rate, µ, of REF and PEP in comparison to the theoretical µ (th. µ) (**C**), and comparison of percentual cell lysis of REF and PEP (**D**). Scans of SDS-PAGE gels are provided in the Appendix A. Peptide content and estimated cell lysis (DNA) were analysed, at least in technical duplicates. Dry cell mass was estimated in a single determination. Theoretical DCM was calculated according to an exponential feed profile with a theoretical specific growth rate, µ, of 0.1 h^−1^.

**Figure 3 ijms-24-02641-f003:**
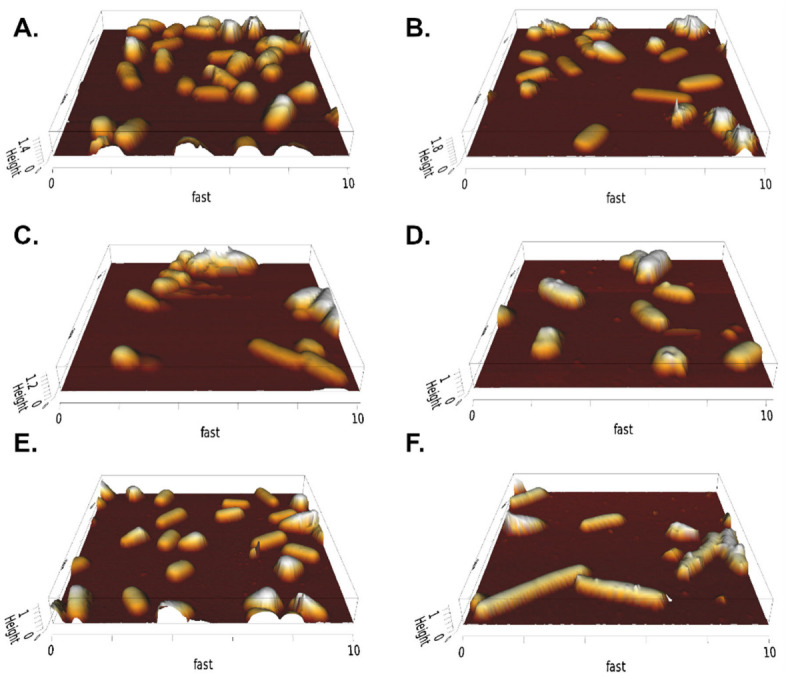
AFM imaging of bacterial cells immobilized on PEI-coated glass slides in PBS. The left column shows REF with growth times of (**A**) 3 h, (**C**) 19 h, and (**E**) 31 h, while the right column shows the PEP with growth times of (**B**) 3 h, (**D**) 19 h, and (**F**) 31 h.

**Figure 4 ijms-24-02641-f004:**
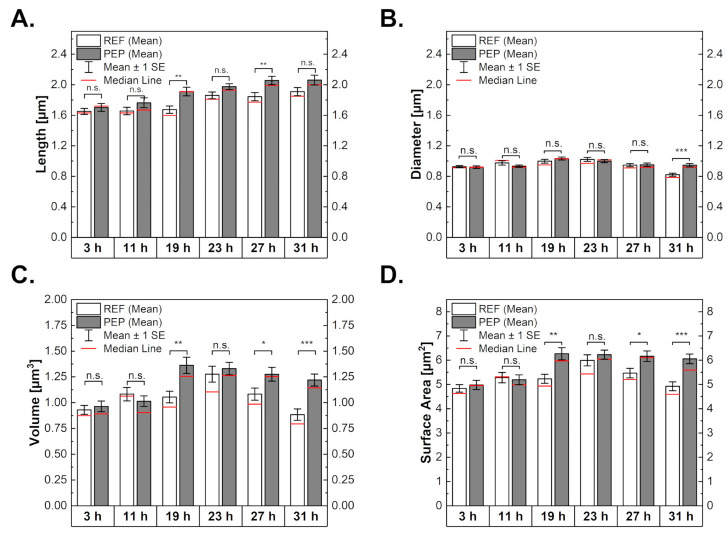
Morphological parameters of bacterial cells for reference (white bars) and peptide-producing cells (grey bars). The figure shows measured lengths (**A**), diameters (**B**), and calculated volumes (**C**), as well as surface areas (**D**). The number of measured cells over 3 to 31 h for the reference sample was 82, 45, 42, 44, 55, and 54, and for the peptide-producing one, 63, 70, 60, 92, 75, and 74. N.s. indicates non-statistically significant differences, * significance with *p*-value < 0.05, ** with *p*-value < 0.01 and *** with *p*-value < 0.001.

**Figure 5 ijms-24-02641-f005:**
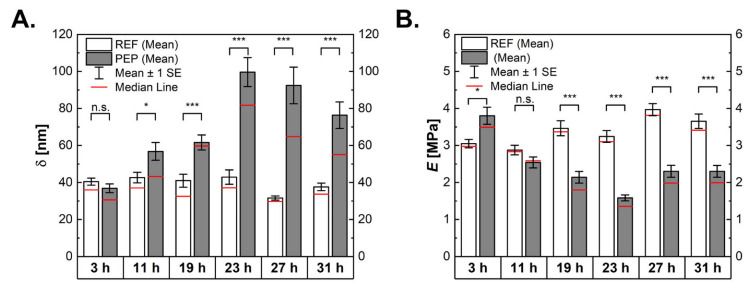
Elastic properties of bacteria shown as indentation (**A**) and apparent Young’s modulus (**B**). The number of measured cells over 3 to 31 h for the reference cells was 76, 55, 42, 51, 51 and 54 and for the peptide-producing cells it was 54, 51, 50, 75, 60 and 57. N.s. indicates non-statistically significant differences, * significance with with *p*-value < 0.01 and *** with *p*-value < 0.001.

**Figure 6 ijms-24-02641-f006:**
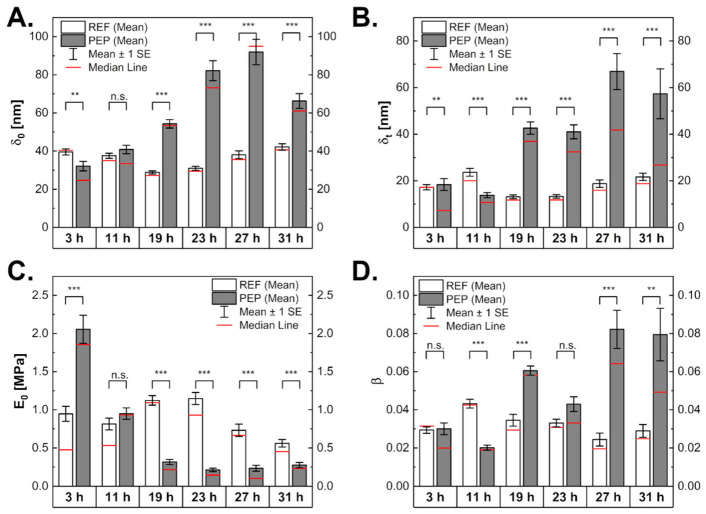
Viscoelastic properties of bacteria from creep experiments shown as initial deformation (**A**), creep magnitude (**B**), relaxation modulus (**C**), and power law exponent (**D**). The number of measurements over 3 to 31 h for the reference cells was 74, 83, 58, 66, 26, and 43, and for the peptide-producing cells, it was 77, 65, 78, 68, 55, and 42. N.s. indicates non-statistically significant differences, ** with *p*-value < 0.01 and *** with *p*-value < 0.001.

## Data Availability

Raw data available upon reasonable request to the corresponding author.

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
