# Peer review of "Recombinant Peptide Production Softens Escherichia coli Cells and Increases Their Size during C-Limited Fed-Batch Cultivation"

_ijms, 2023, doi:10.3390/ijms24032641_

Round 1
Reviewer 1 Report
This study investigated the mechanical properties of Escherichia coli cells under a defined tightly controlled fed-batch cultivation condition, with carbon supply as the limiting factor. The study found that the cells elongated over the cultivation process regardless of the expression of the protein of interest, while the protein-overexpressing cells exhibited significantly increased cell volumes and surface areas. The protein-overexpressing cells also became more elastic and softer than the control cells. While the results are interesting, I have the following concerns.
1. The study claimed the fed-batch culture was C-limited, but there is no data to support that carbon supply was limited in the cultivation process. The glucose content in the culture media should be monitored over the time course, and be included in the manuscript (as supplementary information).
2. The study involved using two recombinant proteins, but it seems the Fab strain was used in experiments for Figure 1, whereas the CASPON-SST strain was used in all the rest experiments? Ideally, both should have been investigated in the freeze-thaw effect experiments and in the subsequent study of mechanical property experiments so that the conclusions would be much stronger. For instance, the softening effect is unclear. It could be due to either the effect from the CASPON-SST protein itself or could be due to expression of any non-functional protein.
3. Was there only one biological replicate for data included in Figure 2? The authors put a lot of words to describe/discuss the results and potential underlying mechanisms, I am concerned about the reproducibility of the results.
4. In Figure 2D, the REF should be included for cell lysis analysis.
Author Response
We want to thank this reviewer for the insightful comments and questions. We have added sections to the supplementary information, added some additional explanations in manuscript and tried to discuss all open questions in this document.
Review report 1
This study investigated the mechanical properties of Escherichia coli cells under a defined tightly controlled fed-batch cultivation condition, with carbon supply as the limiting factor. The study found that the cells elongated over the cultivation process regardless of the expression of the protein of interest, while the protein-overexpressing cells exhibited significantly increased cell volumes and surface areas. The protein-overexpressing cells also became more elastic and softer than the control cells. While the results are interesting, I have the following concerns.
- The study claimed the fed-batch culture was C-limited, but there is no data to support that carbon supply was limited in the cultivation process. The glucose content in the culture media should be monitored over the time course and be included in the manuscript (as supplementary information).
Answer: The two strains are very well characterized in terms of glucose utilization under conditions described. The exponential feed that controls the growth rate assumes constant glucose biomass yield coefficient throughout the cultivation. In Fig 1A the dashed line shows the theoretical biomass that can be produced with a certain amount of glucose. For the reference (REF) the measured biomass values are more or less exactly on that curve. Consequently, all glucose is consumed by the cells.
For the experiments with the peptide producing strain (PEP) there is a deviation of the biomass in the process from the theoretical course but based on results published in DOI: 10.1186/1475-2859-12-58 one can conclude that glucose yield coefficient decreases under production conditions - glucose is then mainly converted into CO2 and by product acetate. Based on the small deviations from the target biomass shown as a dashed line in Fig 1A, it can be assumed that cells grow under strict C-limitation. We have added the following comment and information to the manuscript:
“The influence of product formation on biomass increase and glucose consumption under similar cultivation conditions has been characterized in a detailed C-balancing studies [49]. Based on these results the deviation of the achieved biomass in the PEP experiment from the theoretical value can be attributed to shifts in metabolism leading to increased CO2 and acetate formation.”
These results can be found in reference 49.
- The study involved using two recombinant proteins, but it seems the Fab strain was used in experiments for Figure 1, whereas the CASPON-SST strain was used in all the rest experiments? Ideally, both should have been investigated in the freeze-thaw effect experiments and in the subsequent study of mechanical property experiments so that the conclusions would be much stronger. For instance, the softening effect is unclear. It could be due to either the effect from the CASPON-SST protein itself or could be due to expression of any non-functional protein.
Answer: We want to thank the reviewer for this important comment. The first section (including Figure 1) was dedicated to method development for AFM measurements on bacteria both from batch and fed-batch cultures. Respective experiments were carried out prior to the described fed-batch cultivations shown in the latter sections of the manuscript.
The focus of our study was to investigate changes in mechanical properties over time under tightly controlled conditions in bioreactors independent of the nature of the expressed protein. However, we do acknowledge a possible influence of different protein types on bacterial mechanics. We know that individual effects of different proteins on the host cell are difficult to predict. In the discussion we mention possible adverse effects of the peptide, such as the highly negative charge of the CASPON tag on the outer membrane that can result in changes of mechanical properties. Without further testing, we presumed that CASPON-SST would not alter the cell mechanical properties if frozen/thawed.
- Was there only one biological replicate for data included in Figure 2? The authors put a lot of words to describe/discuss the results and potential underlying mechanisms, I am concerned about the reproducibility of the results.
Answer: We thank the reviewer for this comment. Yes, there was only one biological replicate for the data included in Figure 2. We acknowledge the reviewers concern regarding reproducibility of our experiments. Given the short time for additional experiments/analytics, we try to confirm reproducibility of our experiments with additional data for growth curves and cell lysis. The replicate cultivation with PEP gave almost identical biomass concentrations throughout the cultivation. Cell lysis appeared to be elevated for the duplicate. However, this likely resulted from different timepoints of DNA analysis as DNA tends to be instable. Additional data for growth curves and cell lysis were added to the supplementary material (Figure S2).
Fig. caption: Growth curves (A) and cell lysis curves (B) of duplicate cultivations with PEP. Minor deviations in DCM (e.g. after 7 h) likely arise from errors during sample preparation. Cell lysis variation might arise from differences in timepoint of sample preparation and DNA measurement (DNA instability).
- In Figure 2D, the REF should be included for cell lysis analysis.
Answer: The only samples drawn from the cultivation broth of REF were biomass samples for growth analysis and AFM samples for measurements regarding mechanical properties. Unfortunately, we cannot provide information about cell lysis of REF, as we did not store cell-free media samples during the cultivation, and no additional cultivation was performed. Slight cell lysis is expected for REF, however, it would be considered minor in comparison to PEP, as REF is not undergoing stress from recombinant protein production. This can be supported by the observation that almost no deviation from the theoretical growth curve was visible for REF in Figure 2A.

Reviewer 2 Report
The manuscript by Andreas Weber, Martin Gibisch, Monika Cserjan-Puschmann, José L. Toca-Herrera, Gerald Striedner reports comparison of mechanical properties of peptide-producing and non-producing BL21(DE3) cells in a fed-batch cultivation. Atomic Force Microscopy, a well suited technique for this task is used, both as an imaging and manipulation tool.
I consider it a very valuable study. It provides important data and gives a better understanding of behavior of E. coli as a host for all-important production of recombinant proteins. The manuscript is very well written, with comprehensive data presentation and up to date citations. A thorough discussion of observed results is provided, with interesting remarks concerning bacterial cell behavior during recombinant protein expression.
The only minor problem with a manuscript is error in referencing of Supplementary Materials.
The paper is certainly suitable for publication and will be of interest to wide audience of biophysics and life science researchers alike.
Author Response
Answer to the reviewers
We thank the reviewer for the insightful comments and questions. We have added sections to the supplementary information, added some additional explanations in manuscript and tried to discuss all open questions in this document.
Review report 2
The manuscript by Andreas Weber, Martin Gibisch, Monika Cserjan-Puschmann, José L. Toca-Herrera, Gerald Striedner reports comparison of mechanical properties of peptide-producing and non-producing BL21(DE3) cells in a fed-batch cultivation. Atomic Force Microscopy, a well suited technique for this task is used, both as an imaging and manipulation tool.
I consider it a very valuable study. It provides important data and gives a better understanding of behavior of E. coli as a host for all-important production of recombinant proteins. The manuscript is very well written, with comprehensive data presentation and up to date citations. A thorough discussion of observed results is provided, with interesting remarks concerning bacterial cell behavior during recombinant protein expression.
The only minor problem with a manuscript is error in referencing of Supplementary Materials.
The paper is certainly suitable for publication and will be of interest to wide audience of biophysics and life science researchers alike.
Answer: We are grateful to the reviewer for the support our manuscript has received. We have looked into the Supplementary Materials referencing and corrected the mentioned error.

Reviewer 3 Report
- Please explain the ‘error reference code”?? spread throughout the manuscript's text.
- What happens if inclusion bodies are formed? The example shown here does not constitute an inclusion body. Can an inclusion body example be included in this study to make it more comprehensive? as in most cases, bacterial cells make inclusion bodies.
- What is the effect of leaky expression of the protein on this study?
- How significant is the recombinant protein expression in these observations? How prudent could it be to generalize all cell cultures based on these results? Could this be an effect of this protein on cell biology that makes it fluid? More examples are needed to conclude these results.
- Microscopy data is also required to test the phenotypical changes in the wall and the plasma membrane.
- Are there any changes in the genomic contents of the cell?
- Are there any changes in the lipid constitution of the plasma membrane that causes these changes? Please explain why.
Author Response
Answer to the reviewers
We want to thank the reviewer for the insightful comments and questions. We have added sections to the supplementary information, added some additional explanations in the manuscript and tried to discuss all open questions in this document.
Review Report 3
Please explain the ‘error reference code”?? spread throughout the manuscript's text.
Answer: The error was occurring due to a faulty referencing in the word document, we have now changed this.
What happens if inclusion bodies are formed? The example shown here does not constitute an inclusion body. Can an inclusion body example be included in this study to make it more comprehensive? as in most cases, bacterial cells make inclusion bodies.
Answer: We want to thank the reviewer for this interesting question. It is true that many different proteins are produced in inclusion bodies and inclusion bodies even provide a viable option for downstream processing units. In the present manuscript, we have decided to focus solely on soluble proteins. Due to time limitations in the present study, we are not able to include any data on proteins produced in inclusion bodies. The reviewer’s comment is thought provoking non the less, and it will be interesting to find out in future work if inclusion bodies due influence the mechanical and structural properties of the host organism.
What is the effect of leaky expression of the protein on this study?
Answer: We did not observe any significant leaky expression of the protein.
How significant is the recombinant protein expression in these observations? How prudent could it be to generalize all cell cultures based on these results? Could this be an effect of this protein on cell biology that makes it fluid? More examples are needed to conclude these results.
Answer: The reviewer raises a very important (and not easily answerable) question. In the present manuscript, we observe two different mechanisms that influence the mechanics of the studied cells, the first being the fed-batch cultivation over time and the second the induced expression of a foreign protein. We do not think that the present result can be generalized for all other proteins, and culturing techniques. It will take a widespread effort to investigate this for different proteins, model systems and organisms. The effect on the mechanics (and the metabolic burden) will probably depend on the protein properties such as charge, size, folding mechanisms and more.
We have performed some preliminary measurements of protein expression in the same host in batch culture and saw that expression of three different small peptides leads to significant softening of the cells, while expression of GFP did not change the mechanics of the cells. We initially decided to not include this preliminary data into the present study, as we wanted the focus to be about a novel method (mechanics, viscoelasticity, power laws). We decided to add a new figure to the supplement that shows the apparent Young’s Modulus for these measurements. We hope this satisfies the questions of the reviewer.
A proper investigation of different types of proteins will, in our opinion, make a lot of sense, but then the study should be performed in a different way (either just in batch culture or in fed-batch with fewer data time points).
Microscopy data is also required to test the phenotypical changes in the wall and the plasma membrane.
Answer: The focus of this manuscript were the changes in mechanics of the E. coli cells due to metabolic burden in the fed-batch culture during protein expression. We believe that proper microscopy data (being both nanometric resolution AFM imaging, electron microscopy and superresolution fluorescence microscopy) will warrant an own manuscript. We think that the reviewers comment is very important though, as proper microscopic analysis of host organisms in bioprocesses is often missing. Due to the time limitation (and that there are no residual fed-batch samples left in storage), we are not able to provide further microscopic data.
Nevertheless, we provide some AFM images in Figure 2, that show partly changes in the surface structure of the cells.
Are there any changes in the genomic contents of the cell?
Answer: There are no changes in the genomic contents of the cell. Genome integration of the target proteins was performed by homologous recombination and we would not expect any off-target sequencing. The constructs used were the same as in (https://onlinelibrary.wiley.com/doi/full/10.1002/biot.201800637) where we performed sequencing analysis and saw that there were no other changes in the genome.
Are there any changes in the lipid constitution of the plasma membrane that causes these changes? Please explain why.
Answer: We want to thank the reviewer for this interesting question. We would not expect any changes in the lipid constitution of the plasma membrane (or the outer membrane). We did not perform any analysis of the lipid composition. For gram-negative bacterial mechanics, the two structures that mostly define mechanical properties are the outer membrane and the peptidoglycan. Therefore, we believe that changes in the inner membrane will not significantly impact bacterial mechanics, as also discussed in (https://pubmed.ncbi.nlm.nih.gov/28666084/).

Reviewer 4 Report
Recombinant peptide production softens Escherichia coli cells and increases their size during C-limited fed-batch cultivation
Weber et al
This manuscript examines a potential link between mechanical properties (at the single-cell level) of Escherichia coli grown in carbon limiting conditions, for both cells producing recombinant peptide/protein and control cells.
The conclusions as drawn by the authors are both highly interesting and very significant. However, serious questions arise from the data presented.
During fermentation cell lysis can be a very significant issue, resulting in a heterogeneous mix of lysed and non-lysed cells and cells which have lost the OM. Some of the lysed cell membranes (IM or OM) may reform, generating a wide mix of species. This is a major potential problem for the interpretation of single particle analysis undertaken by the authors. Such analysis needs to be performed on cultures in which the vast majority of cells are unlysed. In addition, heterogeneity in sample populations may arise from plasmid loss and this should be examined/controlled.
The authors do two tests to examine for the degree of cell lysis. The first is based on extracellular [DNA]. This is an extremely problematic method as it assumes no degradation of the released DNA occurs. This is clearly not the case from the data presented in Fig 1D (as mentioned by the authors e.g. in the legend to Figure S2). The second is based on SDS-PAGE analysis. This data is shown in Figure S1. The quality of the figure is extremely poor. No indication is given of what panels A-C represent, some time points have two replicate samples but no corresponding sample e.g. panel C has duplicates of IC 27hr, but no EC 27 hr. More importantly the gels indicate that there are more cellular proteins in the extracellular content than in the intracellular content at multiple time points. Such a very high degree of cell lysis raises very serious concerns about the rest of the analysis i.e. are we simply looking at the effects of cell lysis or lysis of the OM in the rest of the figures? This possibility is raised by the correlation between the first time point for significant cell lysis from S1 being the same time point at which significant differences are seen between REF and PEP in figures4C, 5 and 6A. In addition, there appears to be poor correlation between the data shown in Figure S1 and Figure 2B which is supposed to be quantification of S1 e.g. S1A at 23hrs EC levels of PEP > IC levels (2B has the opposite) and similarly for 27 hrs (S1C vs 2B) and 31 hrs (S1C vs 2B). Furthermore, S1 implies significant differences in replicate samples which do not appear to correlate with the magnitude of the error bars in Fig 2.
A technical question arises from comparing figure 3 with figure 4. Specifically, the heterogeneity in figure 3 appears to be far larger than the standard deviations in figure 4. Was some other filter applied for which cells/particles to measure, if so what was it?
Another (trivial but important) technical question arises from Figure S3, specifically how were the cell length/width and radius of the “half-sphere” at the end of the cell determined i.e. what were the cut offs used? Taking zero as the baseline the width in Figure S3 is just over 1.1um and the length about 1.7um. We are not told in the legend to figure S3 what sample the cells represent, but the length is consistent with the typical data in fig 4A (though without the radius of the “half-spheres), but the diameter is significantly more than that shown in Fig 4B. This implies that zero was not used as the baseline for defining length and width, so what was and how was the radius of the “half-sphere” determined?
With both the issue of cell lysis open and technical questions open, it is difficult to evaluate the remainder of the manuscript except to comment that i) if the authors wish to open their discussion saying that cell morphology changes “due to carbon limitation” they should show evidence of it; ii) they should provide experimental details behind figure S6 e.g. constructs used and discuss more the results in figure S6 vs figure 1; iii) explain why they are significant differences in Fig 6C at 3hrs between REF and PEP i.e. before induction of the peptide and whether this implies there should be biological replicates and not just technical replicates.
Author Response
REFEREE: This manuscript examines a potential link between mechanical properties (at the single-cell level) of Escherichia coli grown in carbon limiting conditions, for both cells producing recombinant peptide/protein and control cells.
The conclusions as drawn by the authors are both highly interesting and very significant. However, serious questions arise from the data presented.
During fermentation cell lysis can be a very significant issue, resulting in a heterogeneous mix of lysed and non-lysed cells and cells which have lost the OM. Some of the lysed cell membranes (IM or OM) may reform, generating a wide mix of species. This is a major potential problem for the interpretation of single particle analysis undertaken by the authors. Such analysis needs to be performed on cultures in which the vast majority of cells are unlysed. In addition, heterogeneity in sample populations may arise from plasmid loss and this should be examined/controlled.
ANSWER: We thank the reviewer for the careful examination and suggestions on how to enhance the clarity and significance of our manuscript. We have considered and answered the comments to our best knowledge. The manuscript was adapted accordingly. We agree that IM/OM aggregates could form from lysed cells, as every culture will most likely contain a certain number of lysed cells and therefore potential newly formed IM/OM aggregates. However, the possibility of mistaking such vesicles with actual cells is rather low, as these vesicles are likely very unstable (very low Young’s Modulus or destroyed during measurement), and vastly outnumbered by actual cells. Such aggregates would therefore represent clear outliers. However, we agree with the reviewer that cell lysis during fermentation can be a very important problem, especially for cells producing recombinant proteins. Yet, in our opinion, it is not a problem in our case since less than 5% of the reference strain lysed, and no more than 20% of the cells were lysed in the peptide-producing strain (DNA as indicator for cell lysis is discussed below). Regarding the possible plasmid loss, we would like to note that the recombinant expression systems used here are genome-integrated and not plasmid-based.
REFEREE: The authors do two tests to examine for the degree of cell lysis. The first is based on extracellular [DNA]. This is an extremely problematic method as it assumes no degradation of the released DNA occurs. This is clearly not the case from the data presented in Fig 1D (as mentioned by the authors e.g. in the legend to Figure S2).
ANSWER: We value the reviewer’s statement and agree that detecting the extracellular DNA is not a perfect method, but a valuable key factor indicating the cell status nonetheless. DNA degradation in the fermentation broth can happen to a certain extent during fermentation, sample preparation, and concomitant analytical procedures and will most likely influence the method. To counteract this during sampling, we reduce the activity of DNases by immediately cooling the samples to 4 °C and later store at them -20 °C. However, this methodology is currently the best possible way for us to approximate cell lysis of our cultivations and was used previously (https://doi.org/10.1002/biot.202000562, https://doi.org/10.1002/btpr.2292). We added the aforementioned references to the respective section regarding DNA analysis (page 3, lane 141). In Figure S2, we want to show the chronological course of cell lysis based on DNA measurements in respect to the growth curves.
REFEREE: The second is based on SDS-PAGE analysis. This data is shown in Figure S1. The quality of the figure is extremely poor. No indication is given of what panels A-C represent, some time points have two replicate samples but no corresponding sample e.g. panel C has duplicates of IC 27hr, but no EC 27 hr.
ANSWER: We apologize for the poor quality of Figure S1 and have adapted it accordingly. Inconsistencies regarding replicates derived from inadequate descriptions in the figure itself.
REFEREE: More importantly the gels indicate that there are more cellular proteins in the extracellular content than in the intracellular content at multiple time points. Such a very high degree of cell lysis raises very serious concerns about the rest of the analysis i.e. are we simply looking at the effects of cell lysis or lysis of the OM in the rest of the figures? This possibility is raised by the correlation between the first time point for significant cell lysis from S1 being the same time point at which significant differences are seen between REF and PEP in figures4C, 5 and 6A. In addition, there appears to be poor correlation between the data shown in Figure S1 and Figure 2B which is supposed to be quantification of S1 e.g. S1A at 23hrs EC levels of PEP > IC levels (2B has the opposite) and similarly for 27 hrs (S1C vs 2B) and 31 hrs (S1C vs 2B). Furthermore, S1 implies significant differences in replicate samples which do not appear to correlate with the magnitude of the error bars in Fig 2.
ANSWER: We thank the reviewer for this important comment. Samples analysed via SDS-PAGE for peptide quantification (Figure S1) were diluted individually to fit the linear range of the BSA standards. The extracellular protein contents in the respective lanes therefore do not represent lysed cells when compared to the lanes that contain intracellular samples. This also led to the impression that extracellular samples contain higher peptide concentrations than the intracellular samples and other misconceptions. We apologize for the missing information regarding the individual dilutions and have updated the Figure S1 accordingly.
REFEREE: A technical question arises from comparing figure 3 with figure 4. Specifically, the heterogeneity in figure 3 appears to be far larger than the standard deviations in figure 4. Was some other filter applied for which cells/particles to measure, if so what was it?
ANSWER: Only cells that showed a cylindrical appearance capped with half-spheres were used for analysis of morphology of the bacteria, no further filter was applied for these measures. The data in Figure 4 is comprised of at least 5 20x20 µm2 AFM images, while the data shown in Figure 3 are meant to show examples of different cell shapes with the fermentation conditions. In addition, Figure 4 uses the standard error of the mean value as error measure.
REFEREE: Another (trivial but important) technical question arises from Figure S3, specifically how were the cell length/width and radius of the “half-sphere” at the end of the cell determined i.e. what were the cut offs used? Taking zero as the baseline the width in Figure S3 is just over 1.1um and the length about 1.7um. We are not told in the legend to figure S3 what sample the cells represent, but the length is consistent with the typical data in fig 4A (though without the radius of the “half-spheres), but the diameter is significantly more than that shown in Fig 4B. This implies that zero was not used as the baseline for defining length and width, so what was and how was the radius of the “half-sphere” determined?
ANSWER: The cell shown in Figure S3 (which is now Figure S4) was indeed one from the standard fermentation conditions sampled at 3 h. Both the cell diameter and the cell length were defined from the 0 µm baseline. The baseline itself was defined by plane fitting and gaussian smoothing. When preparing this figure we wanted to use an image of a single bacteria with high enough resolution and good appearance (meaning no too high noise, sharp edges), the large diameter is within the statistical range of the dataset.
REFEREE: With both the issue of cell lysis open and technical questions open, it is difficult to evaluate the remainder of the manuscript except to comment that
- i) if the authors wish to open their discussion saying that cell morphology changes “due to carbon limitation” they should show evidence of it;
ANSWER: This is a valid concern. We therefore performed additional measurements for glucose analysis via HPLC and added the results to the supplementary material as Figure S3.
The following phrases were included in the manuscript (page 6, line 254-260): “Residual glucose was exemplarily analysed for PEP using HPLC to confirm carbon limitation (data supplemented in Figure S3). The residual glucose content specific to the respective biomass found to be 2.2% on average in the course of cultivation, which corresponds to 1.4% of fed glucose at respective timepoints. A minor accumulation was detected, but this was considered negligible.”
REFEREE: ii) they should provide experimental details behind figure S6 e.g. constructs used and discuss more the results in figure S6 vs figure 1; iii) explain why they are significant differences in Fig 6C at 3hrs between REF and PEP i.e. before induction of the peptide and whether this implies there should be biological replicates and not just technical replicates.
ANSWER: Information regarding the different constructs was added to the caption of Figure S7 (which was before Figure S6). These measurements were performed on batch cultures using the same measurement settings as defined in the materials & methods section. The difference between <oFTN> in Figure S7 and Figure 1 lie within the variance. The reduced modulus for samples producing <oCASP-SST> appears to stem from an increased metabolic burden, but we do not have any further evidence on this yet.
The differences in mechanics between REF and PEP at 3 hours before induction of the peptide production as shown in Figure 6C was puzzling to us as well. As can be seen for both REF and PEP the values show skewed normal distributions and an increase in the number of measurements (and, as the reviewer suggests using biological measurements) would help in determining if this changes just arise from the statistical distributions of the sample properties.

Round 2
Reviewer 1 Report
I still have concerns on my question 2 and 4.
2. Young's Modulus comparison between the freshly harvested and frozen/thawed CASP-SST strain cells should be conducted, so that the impact of freeze/thaw on the mechanic property of these cells could be ruled out.
4. The BL21(DE3) cells without CASP-SST expression cassettes should be included in Figure 2D to show the impact of CASP-SST overexpression on the cell lysis. Also, DNA and peptide contents are affected by both biosynthesis and biodegradation. Therefore, comparing extracellular DNA and peptide percentages could not be used to indicate peptide secretion, and there is no sufficient evidence to demonstrate that the recombinant peptide was leaked out of cells before cell lysis. However, assuming that the nuclease and protease activities are the same with or without overexpression of CASP-SST, relatively increased cell lysis in the recombinant strain (indicated by increased extracellular DNA content) could imply that the softened cell walls of CASP-SST cells might be associated with the production of the recombinant peptide.
Author Response
Answer to the reviewer.
Reviewer 1 comments
I still have concerns on my question 2 and 4.
- Young's Modulus comparison between the freshly harvested and frozen/thawed CASP-SST strain cells should be conducted, so that the impact of freeze/thaw on the mechanic property of these cells could be ruled out.
We performed additional AFM measurements using cells derived from shake flask cultivations (as described) with the PEP strain expressing CASP-SST. Both, freshly harvested and frozen/thawed cells were used for AFM measurements. The results can now be found in the adapted Figure 1. We show that there is no significant difference in the Young’s Modulus of either fresh or frozen cells. The PEP strain behaves softer than the other tested strains, as expected from the results of our fed-batch cultivation.
- The BL21(DE3) cells without CASP-SST expression cassettes should be included in Figure 2D to show the impact of CASP-SST overexpression on the cell lysis. Also, DNA and peptide contents are affected by both biosynthesis and biodegradation. Therefore, comparing extracellular DNA and peptide percentages could not be used to indicate peptide secretion, and there is no sufficient evidence to demonstrate that the recombinant peptide was leaked out of cells before cell lysis. However, assuming that the nuclease and protease activities are the same with or without overexpression of CASP-SST, relatively increased cell lysis in the recombinant strain (indicated by increased extracellular DNA content) could imply that the softened cell walls of CASP-SST cells might be associated with the production of the recombinant peptide.
We thank reviewer 1 for the valuable input. The manuscript was adapted accordingly. Data on cell lysis of the reference strain was gathered and can now be found in an adapted Figure 2D. As shown, REF lysed less extensively in comparison to PEP, and cell lysis levels remained almost constant throughout the cultivation. We can therefore confirm that increased cell lysis levels of PEP derived from expression of the recombinant peptide. Statements on release of CASP-SST into the cultivation medium independent of cell lysis were removed, as these were indeed not sustainable in respect to the explanation kindly provided by reviewer 1. Moreover, we can now strengthen our hypothesis regarding the impact of CASP-SST expression on cell softness and lysis. PEP lysed to a greater extent than REF, and CASP-SST was shown to soften the cells independent of freezing/thawing.

Reviewer 3 Report
I believe more data is required for convincing answers to my questions. I do not think the manuscript is suitable for publication in its present form.
Author Response
The third version of the manuscript contains new measurements and changes to the text.
Reviewer 4 Report
Manuscript has been appropriately corrected
Author Response
We have revised the English as the evaluator, and the editor, have suggested.
Round 3
Reviewer 1 Report
All my concerns have been addressed.
Author Response
We thank the evaluator for his positive response. We have revised the language.